# Comparative analysis of mRNA and protein degradation in prostate tissues indicates high stability of proteins

Wenguang Shao [1,17], Tiannan Guo [1,2,3,17], Nora C. Toussaint [4,5], Peng Xue[1,6], Ulrich Wagner[7], Li Li[8], Konstantina Charmpi[8], Yi Zhu[1,2,3], Jianmin Wu [9], Marija Buljan[1], Rui Sun[2,3], Dorothea Rutishauser[7], Thomas Hermanns[10], Christian Daniel Fankhauser [10], Cedric Poyet[10], Jelena Ljubicic[7], Niels Rupp[7], Jan H. Rüschoff[7], Qing Zhong[7,11], Andreas Beyer [8], Jiafu Ji[12], Ben C. Collins [1], Yansheng Liu [13], Gunnar Rätsch[5,14], Peter J. Wild[7,15] & Ruedi Aebersold [1,16]

Deterioration of biomolecules in clinical tissues is an inevitable pre-analytical process, which affects molecular measurements and thus potentially confounds conclusions from cohort analyses. Here, we investigate the degradation of mRNA and protein in 68 pairs of adjacent prostate tissue samples using RNA-Seq and SWATH mass spectrometry, respectively. To objectively quantify the extent of protein degradation, we develop a numerical score, the Proteome Integrity Number (PIN), that faithfully measures the degree of protein degradation. Our results indicate that protein degradation only affects 5.9% of the samples tested and shows negligible correlation with mRNA degradation in the adjacent samples. These findings are confirmed by independent analyses on additional clinical sample cohorts and across different mass spectrometric methods. Overall, the data show that the majority of samples tested are not compromised by protein degradation, and establish the PIN score as a generic and accurate indicator of sample quality for proteomic analyses.

[1] Department of Biology, Institute of Molecular Systems Biology, ETH Zurich, Zurich 8093, Switzerland. [2] School of Life Sciences, Westlake University, 18 Shilongshan Road, Hangzhou 310024 Zhejiang Province, China. [3] Institute of Basic Medical Sciences, Westlake Institute for Advanced Study, 18 Shilongshan Road, Hangzhou 310024 Zhejiang Province, China. [4] NEXUS Personalized Health Technologies, ETH Zurich, Zurich 8093, Switzerland. [5] SIB Swiss Institute of Bioinformatics, Zurich 8093, Switzerland. [6] Institute of Biophysics, Chinese Academy of Sciences, Beijing 100101, China. [7] Institute of Surgical Pathology, University Hospital Zurich, Zurich 8091, Switzerland. [8] CECAD, University of Cologne, Cologne 50931, Germany. [9] Key laboratory of Carcinogenesis and Translational Research, Center for Cancer Bioinformatics, Peking University Cancer Hospital & Institute, Beijing 100091, China. [10] Department of Urology, University of Zurich, University Hospital Zurich, Zurich 8091, Switzerland. [11] Cancer Data Science Group, Children's Medical Research Institute, University of Sydney, Sydney 2050 NSW, Australia. [12] Key laboratory of Carcinogenesis and Translational Research, Department of Gastrointestinal Surgery, Peking University Cancer Hospital & Institute, Beijing 100091, China. [13] Department of Pharmacology, Cancer Biology Institute, Yale University School of Medicine, West Haven, CT 06516, USA. [14] Department of Computer Science, ETH Zurich, Zurich 8093, Switzerland. [15] Dr. Senckenberg Institute of Pathology, University Hospital Frankfurt, Frankfurt am Main 60590, Germany. [16] Faculty of Science, University of Zurich, Zurich 8057, Switzerland. [17]These authors contributed equally: Wenguang Shao, Tiannan Guo. Correspondence and requests for materials should be addressed to T.G. (email: guotiannan@westlake.edu.cn) or to P.J.W. (email: peter.wild@kgu.de) or to R.A. (email: aebersold@imsb.biol.ethz.ch)

Precise and accurate measurements of different types of molecules in tissue samples are essential for clinical diagnosis, prognosis, and therapy. High-throughput methods, including next-generation sequencing[1] and proteomic technologies[2], allow for the measurement of thousands of molecules expressed in tissue samples in different physiological and pathological states, providing an improved understanding of molecular mechanisms of diseases. However, pre-analytical factors such as artefactual degradation, in vitro modification of molecules or loss through leakage may alter qualitative and quantitative molecular patterns in clinical specimens, thus confounding measurements and clinical conclusions.

In living cells, mRNA levels are tightly controlled by balancing the protection of mRNA molecules from unwanted degradation and the directed degradation of specific RNA species by RNA-degrading enzymes including ribonucleases (RNases). In isolated tissue samples, this balance is frequently disrupted, leading to artefactual changes in mRNA profiles that currently can hardly be modeled. In a recent study, Romero and colleagues observed a significant correlation between the storage time of peripheral blood mononuclear cell samples at room temperature and extent of RNA degradation[3]. Importantly, the observed decay rate varied for different transcripts[3]. While it remains controversial whether and how the bias introduced by mRNA decay can be corrected in silico, in most studies samples in which mRNA degradation exceeds an arbitrary threshold, as measured by an objective score, are treated separately or discarded[3,4]. For example, about 21% of prostate tumors analyzed in a recent TCGA study were found to be substantially degraded and were excluded from further analysis[4].

Similarly for proteins, homeostasis is also a complex and tightly controlled process in living cells. Protein degradation is an important component of protein homeostasis. It is catalyzed by proteases and specific protein degradation pathways. In resected or stored tissue samples, additional proteases, including caspases associated with cell death pathways are activated, and the general protein degradation system is altered. Using mass spectrometric measurements of blood samples, a few studies have investigated the effects of sample storage on protein stability. Remarkably, the plasma proteome appeared to be highly stable even after storage for a week at 4 °C or at room temperature in EDTA plasma tubes[5]. However, storage time caused changes in antibody-depleted plasma proteome profiling, probably due to structural alteration of some proteins[5].

Little is known about protein degradation in tissue samples and how it might affect downstream results. An in vitro study showed that the rate of protein degradation in skeletal muscle tissue was affected by incubating the samples with soluble amino acids (i.e., leucine) and chemical elements (i.e., calcium; zinc)[6]. Recent technical advances in mass spectrometry (MS), specifically the development of SWATH/DIA methods have made highly reproducible quantification of thousands of proteins in clinical cohorts consisting of hundreds of samples a reality[7,8]. The question whether protein degradation, if present in clinical samples, affects the quality of proteomic measurements and thus introduces a bias in biological and clinical conclusions is therefore of great and acute importance, particularly for clinical cohort studies.

Here, we comprehensively compare the degradation patterns of mRNA and protein in a prostate cancer tissue cohort. We subject 68 pairs of adjacent tissue samples to RNA-Sequencing (RNA-Seq) or proteomic analysis by pressure cycling technology (PCT) coupled with SWATH mass spectrometry[7]. We develop a score, the Proteome Integrity Number (PIN), to objectively quantify the proteome-wide degree of degradation in each sample. We benchmark the PIN algorithm using a set of ground-truth samples in which the levels of proteome degradation were artificially controlled and independently validated, and assess the relative degree of mRNA and protein degradation in adjacent samples. Our results show that protein degradation, although present and detectable, has a minimal impact on proteomic measurements in the clinical cohort tested, and is relatively independent of mRNA degradation. We also establish the PIN score as an accurate indicator of sample quality for proteomic analyses of clinical samples and show that the PIN score is robust across different types of clinical samples and mass spectrometric measurement methods. Our results thus provide important information and resources for proteomic measurements in clinical cohort studies.

## Results and discussion

**Development and validation of the PIN algorithm.** To estimate the degree of protein degradation in a sample, we first introduced an individual protein integrity score (iPIS) for each measured protein. This score quantifies the relative abundance of semi-tryptic peptides, the likely products of protein degradation by endo- and exo-proteases, in relation to the total number of peptides detected from the protein. To estimate an overall degree of protein degradation on the sample level, the PIN of each sample is then calculated as the arithmetic mean of all iPIS values (Fig. 1). Numerically, a PIN ranges from zero to one. A higher PIN indicates a smaller fraction of protein degradation products and thus a higher degree of proteomic integrity of the sample. Even though the PIN score makes no statement about the extent of degradation of proteins that were not identified in the data set, which for the most part are proteins of low abundance, it nevertheless provides an almost complete picture of protein degradation on the sample level. This is due to the abundance distribution of proteins in cells, which implies that more than 99% of the total protein mass is represented in the top five orders of magnitude of protein abundance, the range approximately covered by mass spectrometric measurements[9]. Another advantage of focusing on proteins with higher abundance is an increased likelihood that semi-tryptic peptides, which usually show lower mass spectrometric signals compared with fully tryptic peptides, can be consistently and reliably quantified (Supplementary Fig. 1a, b).

To validate the PIN algorithm, we performed a benchmarking study (Supplementary Note 1). A set of 'ground truth' samples was generated in which the levels of proteome degradation for each sample were known and calibrated. Specifically, six samples (A-F) of protein extracts of HeLa Kyoto cells were treated with different amounts of the low specificity protease Proteinase K to generate samples of progressively increasing protein degradation, and nine samples were prepared without treatment as controls (Table 1). As an orthogonal validation method, sodium dodecyl sulfate-polyacrylamide gel electrophoresis (SDS-PAGE) was applied to the biological replicates of six treated samples and one control sample. As shown in Fig. 2a, the treated samples demonstrated strong and progressive evidence of proteome degradation as expected—intact proteins breaking into their smaller sub-units, resulting in the presence of a larger number of low molecular weight bands in the gel image and concurrent depletion of the higher molecular weight bands.

We then calculated the PIN score as an indicator of proteome integrity of each sample in the benchmarking datasets, by applying the PIN algorithm to the quantitative peptide matrix obtained from SWATH-MS measurements. Not surprisingly, the PIN values of the treated samples decreased with progressing protein degradation. Interestingly, the PIN values showed a strong linear relationship ($R^2$: 0.94) with the amount (on a logarithmic scale) of Proteinase K added, suggesting that PIN values are accurate indicators of the degree of protein degradation

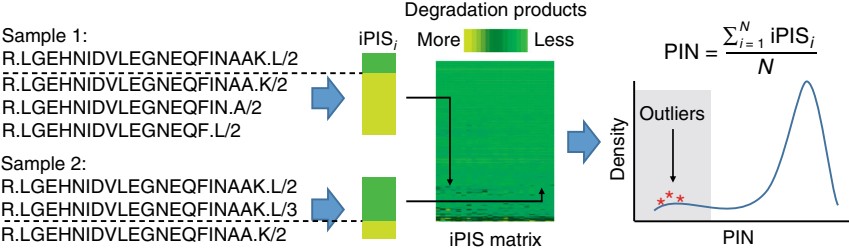

**Fig. 1** The PIN algorithm. Shown is a schematic diagram of the computational workflow

| Table 1 The overall experimental design and results of the benchmarking study | | | | | |
|---|---|---|---|---|---|
| **Sample** | **# biological replicates** | **# technical replicates** | **Proteinase K concentration (μg/μl)** | **PIN** | **P-value** |
| A | 2 | 2 | 0.0200 | 0.645 | 3.17E-17 |
| B | 2 | 2 | 0.0100 | 0.714 | 2.07E-13 |
| C | 2 | 2 | 0.0040 | 0.850 | 2.73E-06 |
| D | 2 | 2 | 0.0020 | 0.868 | 2.15E-05 |
| E | 2 | 2 | 0.0010 | 0.900 | 9.87E-04 |
| F | 2 | 2 | 0.0005 | 0.933 | 0.039 |
| Control | 18 | 2 | N.A. | 0.964 | 0.721 |

P-values were obtained from the PIN algorithm by probability distribution fitting

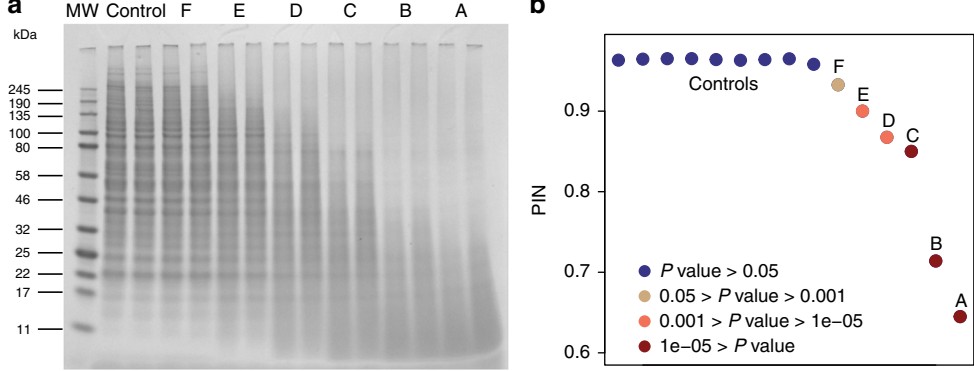

**Fig. 2** The benchmarking study to validate the PIN algorithm. **a** The uncropped SDS-PAGE gel showing the effect of sample treatment with different amounts of protease on the proteome level. Lanes: 1, MW Marker; 2, Controls (without protease treatment); 3–8, Samples F-A as defined in Table 1. **b** Scatter plot of sorted PIN values of the samples of the benchmarking study. The samples with P-values above 0.05 are marked in blue; those with P-values between 0.001 and 0.05 are in coffee; those with P-values between 1e−5 and 0.001 are in orange; those with P-values below 1e−5 are in dark red

(Supplementary Fig. 2). After statistical analysis that estimates a P-value indicating the probability of observing such a specific PIN value under the null hypothesis that the sample was not degraded, we were able to confidently identify the samples with protease treatment (Table 1 and Fig. 2b) with different statistical significances. For example, three samples treated with the highest amounts of protease added were identified as most extensively degraded, with P-values below 1e−5; next, two samples were identified as degraded with P-values between 1e−5 and 0.001; the sample treated with the minimal amount of protease was identified as moderately degraded with a P-value of 0.04 (Fig. 2b). Overall, we showed that PIN values provide precise measurements of proteomics samples to infer protein degradation (Supplementary Figs. 2 and 3) and that the workflow is sensitive and robust to identify degraded samples of various extents with correct statistical significances (Table 1; Fig. 2b; Supplementary Table 1 and see Supplementary Note 1 for details).

**Comparative analysis of transcript and protein degradation.** Next, to further evaluate the degree of mRNA and protein degradation in directly comparable tissue samples, we produced 68 pairs of adjacent tissue punches from resected prostates of 24 prostate cancer patients (Fig. 3a and Supplementary Table 2) and analyzed the degradation of transcripts and proteins, respectively. For each pair, one punch was analyzed using PCT-SWATH, while the adjacent punch was analyzed with RNA-Seq. Overall, we measured 14,593 expressed genes using RNA-Seq. By performing targeted data extraction against an assay library containing ions of 51,969 tryptic peptides and 4530 semi-tryptic peptides, we detected in total 3056 proteins represented by 29,818 peptides, consisting of ions of 27, 685 tryptic peptides and 2133 semi-tryptic peptides.

We assessed the degradation of mRNA in each tissue sample using mRIN, which estimates degradation by quantifying the 3′ bias of each gene. Assuming that the transcriptome of most

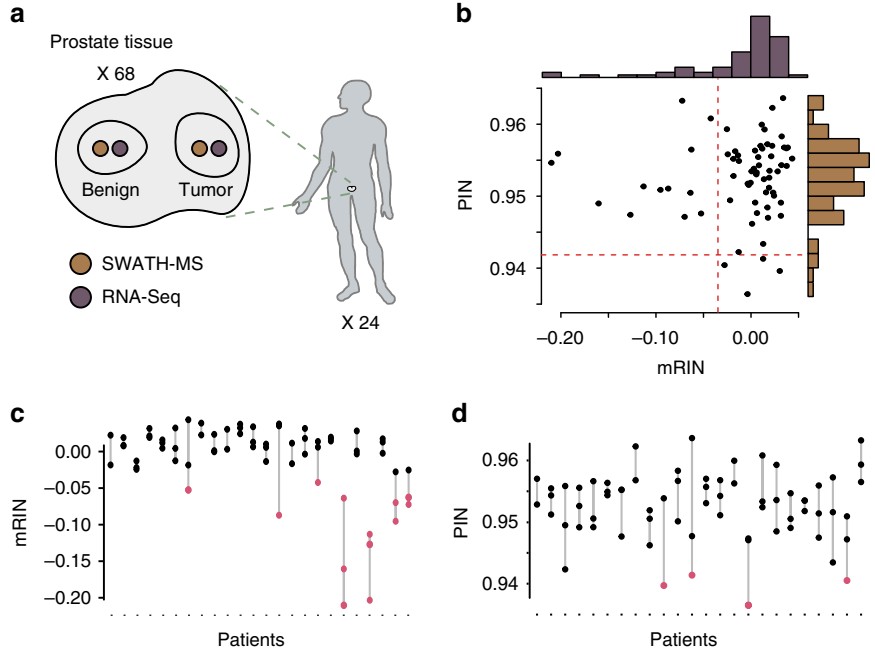

**Fig. 3** Comparisons of mRNA and protein degradation in the clinical cohort. **a** The experimental design. The human figure is a modified version of a drawing from Buljan et al.[31], published under a CC BY 3.0 license. **b** Comparison of the degrees of degradation at mRNA and protein levels, measured by mRIN and PIN respectively ($n = 68$). **c** Patient-specific mRNA degradation patterns observed. Samples with significant mRNA degradation are marked in red. **d** Patient-specific protein degradation patterns observed. Samples with significant protein degradation are marked in red

samples is not degraded, we detected substantial mRNA degradation (mRIN < −0.04; *P*-value < 0.01) in 13 (19.1%) out of 68 prostate tissue samples (Fig. 3b, Table 2 and Supplementary Table 3). This level of significant degradation is in accordance with mRNA degradation observed in a previous study[4]. Here, unsupervised hierarchical clustering of gene expression profiles separated seven samples into a distinct cluster. Of these, five samples exhibited a significant degree (*P*-value < 0.01) of mRNA degradation (Supplementary Fig. 4a). The remaining two samples showed intermediate degradation (*P*-values: 0.017 and 0.045), indicating that mRNA degradation would probably lead to distorted biological conclusions by introducing unanticipated bias.

We then computed the PIN values for the 68 prostate tissue samples to estimate the degree of protein degradation. The PIN values of this set of samples ranged from 0.937 to 0.964 (mean: 0.953; standard deviation: 0.005) (Fig. 3b and Supplementary Table 3). After statistical analyses, we identified four (5.9%) samples with substantial protein degradation (PIN: 0.937, 0.940, 0.941, and 0.941) at a significance level with *P*-values below 0.01 (Table 2). Based on protein expression profiles, three of these samples were clustered within a subgroup of 10 samples in which the other seven samples did not show significant degradation based on the PIN values, suggesting that in this set of samples, protein degradation would not likely bias biological and clinical conclusions by introducing a subgroup of samples with substantial protein degradation (Supplementary Fig. 4b).

Since the tissue samples for RNA-Seq and proteome analysis were from adjacent tissue regions, we assumed an equal degree of pre-analytical variation for each sample pair from a specific tissue. We then asked whether mRNA and proteins were degraded to a similar extent in a specific tissue. Interestingly, none of the samples with substantial protein degradation showed severely degraded mRNA, and vice versa. Further, we observed a negligible correlation between mRNA degradation and protein degradation (Spearman rho: 0.147; *P*-value: 0.232) (Fig. 3b).

Therefore, our results reveal negligible dependency between transcriptome and proteome degradation in this set of samples.

We then compared the degree of degradation at the level of individual transcripts and proteins in this sample cohort. We observed that at the same significance level (*P*-value < 0.01), fewer samples exhibited degradation at the protein than the transcript level. Only 5.9% of the samples displayed statistically significant protein degradation, whereas 19% did at the transcript level. Moreover, the *P*-values of degraded samples of transcripts were much more significant than those of proteins (Table 2), indicating that the degree of mRNA degradation was higher in the sample cohort studied.

Of note, the five samples with the most extensive mRNA degradation were exclusively derived from two patients (Fig. 3c). In contrast, proteomic degradation displayed no bias to patients (Fig. 3d). Moreover, 11 out of 13 samples with significant mRNA degradation were from tumor samples (Table 2), suggesting that mRNA degradation occurs more frequently in tumors compared to benign tissues. One possible explanation for this is the deregulated mRNA degradation machinery associated with tumors[10].

To investigate the stability of each individual transcript and protein, we further analyzed the degradation profiles of transcripts and proteins across the sample cohort using mKS (Fig. 4a) and iPIS matrices (Fig. 4b). Sorted by the degree of degradation measured by mRIN (i.e., 3′ bias) and PIN (i.e., fraction of protein degradation products) on the sample level, samples demonstrated consistently sorted degrees of degradation of individual molecules, confirming the effectiveness of both scoring systems to measure degradation. In general, we observed a higher degree of degradation at the transcriptome level compared to the proteome level, with most proteins exhibiting high stability in all samples, even in those with lower PIN values. In the samples with significant mRNA degradation, most genes were affected, thus indicating universal degradation of transcripts (Fig. 4a and Supplementary Fig. 5a).

**Table 2 mRINs and PINs with their associated _P_-values of the degraded prostate tissues**

| | Patient ID | Tissue type | mRNA | | Protein | |
|---|---|---|---|---|---|---|
| | | | mRIN | _P_-value | PIN | _P_-value |
| mRNA degraded samples | 33 | Tumor | −0.210 | 1.6E−40 | 0.955 | 0.542 |
| | 35 | Tumor | −0.203 | 3.9E−38 | 0.956 | 0.649 |
| | 33 | Tumor | −0.161 | 4.6E−25 | 0.949 | 0.187 |
| | 35 | Benign | −0.127 | 7.9E−17 | 0.947 | 0.133 |
| | 35 | Tumor | −0.113 | 6.2E−14 | 0.951 | 0.303 |
| | 37 | Tumor | −0.096 | 1.2E−10 | 0.951 | 0.276 |
| | 14 | Tumor | −0.087 | 2.6E−09 | 0.951 | 0.286 |
| | 38 | Tumor | −0.073 | 3.6E−07 | 0.963 | 0.997 |
| | 37 | Tumor | −0.070 | 7.9E−07 | 0.947 | 0.125 |
| | 33 | Benign | −0.064 | 4.7E−06 | 0.951 | 0.255 |
| | 38 | Benign | −0.063 | 6.2E−06 | 0.957 | 0.697 |
| | 7 | Tumor | −0.053 | 8.6E−05 | 0.955 | 0.542 |
| | 31 | Tumor | −0.043 | 8.5E−04 | 0.956 | 0.649 |
| Protein degraded samples | 16 | Benign | −0.004 | 0.212 | 0.937 | 9.4E-05 |
| | 10 | Tumor | 0.030 | 0.895 | 0.940 | 0.0015 |
| | 37 | Benign | −0.028 | 0.012 | 0.941 | 0.0028 |
| | 12 | Tumor | 0.013 | 0.576 | 0.941 | 0.0052 |

The degraded tissues were significantly degraded at either transcript or protein level (_P_-value < 0.01). _P_-values were obtained from the mRIN and PIN algorithm, respectively

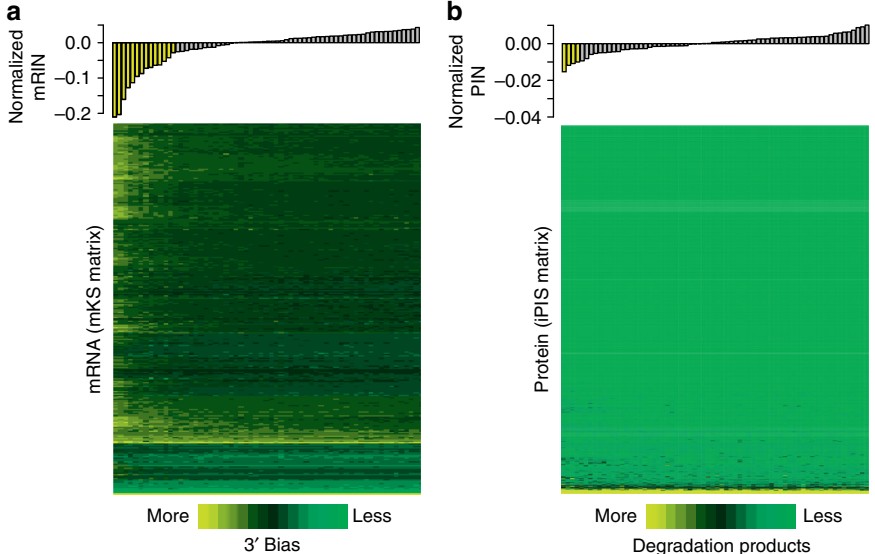

**Fig. 4** Visualization of degradation at the level of individual transcripts and proteins. **a** Heatmap (bottom) of mKS matrix illustrating the degree of degradation of each individual transcript (rows), with samples (columns) ordered by mRINs. The bar plot (top) displays each sample's mRIN (median centered). Samples with significant mRNA degradation are highlighted in yellow in the bar plot ($n = 68$). **b** Similar to **a**, but for iPIS matrix with samples ordered by the PIN values. Four samples with significant protein degradation are highlighted in yellow in the bar plot ($n = 68$)

In contrast, 2384 (78%) proteins with averaged iPIS equal to 1 were detected in all samples, which means that most proteins that we have measured were not affected by protein degradation, even in the samples with lower PIN values, and that protein degradation is more strongly protein specific (Fig. 4b and Supplementary Fig. 5b). Therefore, our results highlight the remarkable difference between universal degradation at the transcriptome level and protein-specific degradation at the proteome level observed in this study. The above results are thus indicative that the pre-analytical variables causing the respective mRNA and protein degradation are decoupled. This could be possibly caused by the difference between the mechanisms of mRNA and protein degradation (mediated by RNase and proteases, respectively). An in-depth elucidation

of degradation mechanisms is beyond the scope of this manuscript.

**The extent of proteomic measurements affected by degradation.** As we confirmed the presence of protein degradation in this set of prostate tissues, we further examined how degradation would affect our proteomic measurements and the clinical conclusions from the study. We defined proteins with averaged iPIS scores below 0.8 (Supplementary Fig. 5b) as degradation-prone proteins in the respective study. To examine whether any bias in biological conclusions would be introduced, we compared the dendrograms of hierarchical clustering using quantitative proteomic data with and without these proteins.

Overall, the two dendrograms were highly similar to each other, with a cophenetic correlation coefficient of 0.94. As shown in Fig. 5a, b, only five samples slightly changed their positions between the dendrograms upon removal of degradation-prone proteins. These observations further suggest that the biological conclusions were not substantially affected by protein degradation. This can be explained as follows. First, in this study, only 149 degradation-prone proteins were detected (Fig. 4b and Supplementary Fig. 5b). These proteins make up 4.9% of the 3056 proteins measured, a fraction of proteins that is likely too small to cause a clustering rearrangement of the total proteome. Second, we used the most intense peptide that could be consistently detected across the sample cohort for protein quantification in the present study. In this way, only of 3.5% proteins on average were quantified by semi-tryptic peptides, which further reduced the effect of protein degradation. This is exemplified by two representative proteins (Supplementary Fig. 1a, b). Although a few semi-tryptic peptides of these two proteins were detected, the most intense and most reproducibly detected peptides were fully tryptic.

In quantitative proteomics, peptides are quantified as surrogates for proteins, and proteins usually generate several peptides that are mass spectrometrically analyzed and can be used for quantification. To reliably quantify abundances of proteins, it is necessary to select the most representative peptides[11] and usually semi-tryptic peptides are not searched as candidates in a routine proteomics data analysis, as the sensitivity of peptide identifications would be likely reduced due to search space expansion. In addition, detection of semi-tryptic peptides was less consistent than that of fully tryptic peptides (Fig. 5c). Further, compared to fully tryptic peptides, a weaker covariation of semi-tryptic peptides with other peptides generated from the same protein was observed (Fig. 5d). These observations suggest that semi-tryptic peptides, albeit indicative of protein degradation extent, should be excluded for protein quantification in routine proteomic experiments[12].

**General utility of the PIN algorithm**. To demonstrate the general utility of the PIN algorithm (Supplementary Note 2), we applied it to additional datasets from clinical proteomics research studies that included various types of clinical samples (specifically prostate tissue, breast tissue, gastric tissue, and human plasma), different sample storage methods (specifically formalin-fixed paraffin-embedded (FFPE) and fresh frozen samples) and different MS instruments, data acquisition and proteomics techniques (specifically data-dependent acquisition (DDA) vs. data-independent acquisition (DIA); iTRAQ labeling vs. label-free quantification; whole samples vs. fractionated samples). The diversity of these studies (Supplementary Table 4) allowed us to demonstrate the general utility of the PIN algorithm across the commonly used MS platforms and sample types and to investigate generic issues related to protein degradation in different (clinical) sample types (Supplementary Figs. 6–12; detailed in Supplementary Note 2). For example, the study of breast cancer tissue samples also confirmed the previous observations reported in[13], as a high degree (91.7%) of overlapped samples was identified by the PIN algorithm and stated in the original report (Supplementary Figs. 6 and 7). Further, the study of gastric cancer tissue samples confirmed the aforementioned observation that the pre-analytical variables causing the respective mRNA and protein degradation are decoupled (Supplementary Figs. 8 and 9).

In summary, our results uncover remarkable stability of proteins in primary clinical tissue specimens, and consolidate the foundation for protein-based clinical diagnosis using mass spectrometry-based bottom-up proteomics. Protein degradation, although present, affected only a relatively limited number of measured proteins, and we only identified four samples with substantial degradation in a clinical cohort of 68 samples. Importantly, protein degradation was relatively independent of mRNA degradation, indicating that pre-analytical variables causing mRNA and protein degradation were decoupled. These findings have important implications for the use of proteomic measurements (e.g., SWATH/DIA MS) in clinical studies, especially for large clinical cohorts, where pre-analytical variables

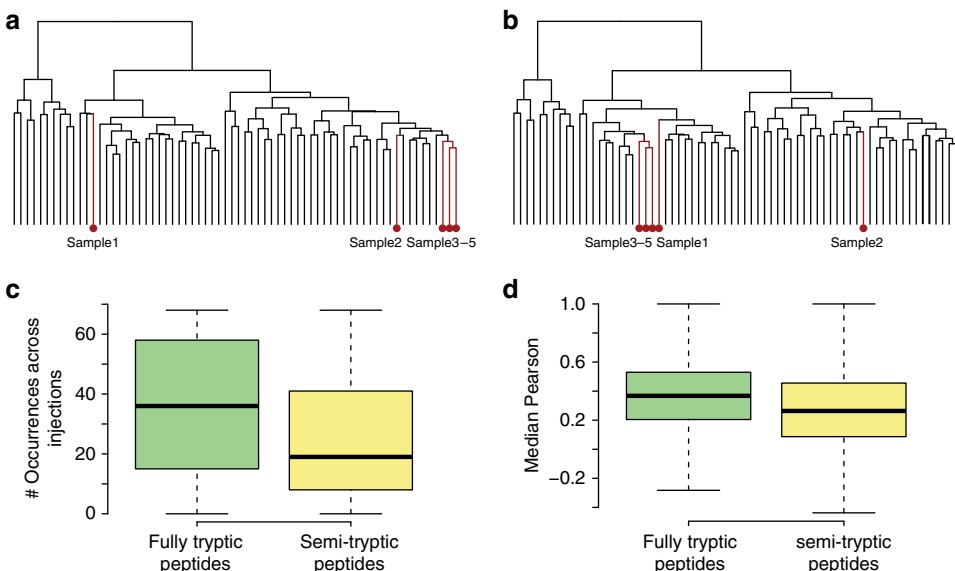

**Fig. 5** Detailed mRNA and protein degradation characteristics. **a** Hierarchical clustering of all proteins quantified in the sample cohort. **b** Hierarchical clustering of quantified proteins without degradation-prone proteins. **c** The reproducibility of fully tryptic peptides was higher than that of semi-tryptic peptides, as demonstrated in the box plot. **d** A stronger covariation of fully tryptic peptides across 68 samples with other peptides generated from the same protein was observed, compared with that of semi-tryptic peptides. The line within the box represents the median, the borders of the box represent the interquartile range and the whiskers represent smallest and largest values no more than 1.5 times the interquartile range; $n = 27{,}685$ for fully tryptic peptides, $n = 2133$ for semi-tryptic peptides

cannot be precisely and equally controlled (Supplementary Note 2). Finally, to identify proteomic samples with substantial degradation, the PIN algorithm (Supplementary Fig. 13 and Supplementary Note 3) developed here is generally applicable to data generated from any bottom-up proteomic datasets by only modifying the search strategy during data analysis without the requirement of any additional experiments. This is especially useful for SWATH/DIA MS datasets, where complete proteome maps, including potential degradation products, are digitally recorded in an unbiased manner.

## Methods

**Prostate cancer tissue specimens.** Sixty-eight prostate tissue samples of 24 patients (Supplementary Table 2) were collected within the ProCOC study[14]. From each patient, a pair of adjacent punches (inner diameter 1 mm) of malignant and benign tissue, respectively, was collected from low-grade prostate tumors. In patients with intermediate and high-grade tumors, we collected pairs of punches from two histologically distinct regions of malignant tissue and a pair of punches of benign tissue. From each pair of tissue punches, one sample was analyzed using RNA-Seq and one by PCT-SWATH. All relevant ethical regulations have been complied. The Cantonal Ethics Committee Zurich (KEK-ZH) has approved all procedures involving human material, and each prostate cancer patient has signed an informed consent form (KEK-ZH-No. 2008-0040).

**RNA-Seq.** RNA sequencing was performed at the Functional Genomics Center Zurich. RNA-Seq libraries were generated using the TruSeq RNA stranded kit with PolyA enrichment (Illumina, San Diego, CA, USA). Libraries were sequenced on an Illumina HiSeq 2500 (Illumina, San Diego, CA, USA) producing an average of 109,198,783 $2 \times 126$ bp paired-end reads per sample. Reads were mapped to the human reference genome (GRCh37) using the STAR aligner (version 2.4.2a)[15]. FeatureCounts[16] was used to determine read counts for all genes annotated in ENSEMBL v75. The mRNA expression profile was generated using log2 transformed transcripts per million (TPM) values, from genes whose averaged TPM values were larger than 4. mRIN[17] was used to estimate the degree of mRNA degradation and P-value of each sample. mRIN uses a modified Kolmogorov–Smirnov (KS) statistic to quantify the 3′ bias per gene. Cohort-normalized KS statistics were stored as mKS matrix, in which columns represent samples and rows represent expressed genes. The mRIN score of an individual sample was defined as the negative mean mKS overall genes in the respective sample.

**PCT-SWATH.** PCT-SWATH analyses[7,18,19] were performed as follows. After washing away O.C.T.®, each tissue punch was lysed and digested using PCT and the PCT-MicroPestle system (Pressure Biosciences Inc., South Easton, MA), followed by C18 cleanup. One microgram of total peptide mass from each sample, as measured by NanoDrop A280, was analyzed in duplicate by SWATH-MS on a 5600 TripleTOF mass spectrometer (Sciex) coupled with a 1D+ Nano LC system (Eksigent, Dublin, CA). A two-hour gradient and 32 fixed SWATH window scheme were adopted as described[7].

**SWATH data analysis.** A SWATH assay library was compiled from 79 data-dependent acquisition (DDA) MS analyses of prostate tissues in a TripleTOF 5600 mass spectrometer. These DDA tandem mass spectra (MS/MS) were searched by Comet[20] and X!Tandem[21] using the default settings with the enzyme set semi-tryptic, to enable the identification of potential degradation products. The search results were validated by PeptideProphet[22] and combined by iProphet[23], and further filtered at a false discovery rate (FDR) of 1% at the peptide level (iProphet probability: 0.7777). The identified spectra were imported into a redundant spectral library by SpectraST[24]. A consensus tandem mass spectral library was then constructed by SpectraST. The generated library was further converted to TraML using the tool ConvertTSVToTraML, with decoy assays appended using the OpenSwathDecoyGenerator.

The SWATH.wiff files were first converted into profile mzXML using msconvert[25]. Through the iPortal workflow manager, the resulting 136 SWATH-MS mzXML files were analyzed by OpenSWATH[26] by default settings as previously described, except that the following parameters were modified: m/z extraction window = 0.05 Thomson; RT extraction window = 600 s. After the targeted extraction of fragment ion chromatograms, pyprophet[27] was used to calculate a single discriminant score from a subset of the scores (library_corr yseries_score xcorr_coelution_weighted massdev_score norm_rt_score library_rmsd bseries_score intensity_score xcorr_coelution log_sn_score isotope_overlap_score massdev_score_weighted xcorr_shape_weighted isotope_correlation_score xcorr_shape) and to estimate the q-value to facilitate FDR control. TRIC[28] was then run on the pyprophet results to perform feature alignment to re-rank peak groups obtained in the original targeted extraction stage with the following parameters (realign_method: spline,

dscore_cutoff: 1, target_fdr: 0.01, max_rt_diff: auto_3mediansdtdev, method: global_best_overall). The identified peptide ions with m_score below 0.000908 (to enable an FDR of 0.01) were then kept and only proteotypic peptides, that are uniquely attributable to one single protein, were kept for the remaining analysis. Replicates were merged for each sample, and proteins that were consistently detected over 50% of the samples were accepted for protein quantification using the most intense peptide.

**The PIN algorithm.** To indicate the stability of each individual protein, we define the iPIS as one minus the ratio of total intensities of semi-tryptic peptides to those of all peptides:

$$\text{iPIS} = 1 - \frac{\sum \text{Intensity}_{semi-tryptic\ peptides}}{\sum \text{Intensity}_{all\ peptides}} \tag{1}$$

Semi-tryptic peptides are peptides that are truncated, possibly by proteases, from one end (either N-terminal or C-terminal) of fully tryptic peptide. Note that intensities were normalized by taking square roots of raw intensities by default in the PIN package (For more option usages and descriptions, see Supplementary Note 3 as the user manual for the PIN algorithm).

To indicate the degradation state of each sample, we define a global Proteome Integrity Number (PIN) as the arithmetic mean of iPISs of all $N$ proteins identified in the sample:

$$\text{PIN} = \frac{\sum_{i=1}^{N} \text{iPIS}_i}{N} \tag{2}$$

Samples showing extensive degradation, i.e., those containing a higher fraction of semi-tryptic peptides have smaller PIN values, analogous to the same trend apparent in the RIN and mRIN scoring theme for RNA. Conveniently, PIN values range from zero (completely degraded samples) to one (samples showing no indication of degradation).

To generate a statistically significant confidence value that can be used as an objective cut-off to reject unsuitable samples, we calculated the P-value of a sample with a specific PIN value that indicates the probability of observing such a specific PIN value under the null hypothesis that the sample was not degraded at the protein level.

To convert PIN values into P-values, we developed a statistical model, similar to that proposed in the mRIN algorithm[17]. Assuming that PIN values of non-degraded samples in a cohort follow a Weibull distribution with the scale parameter $\lambda$ and the shape parameter $k$, the P-value of a specific PIN value $x$ can be then calculated as the tail probability of observing $x$ under the null distribution, that is the cumulative distribution function $P(X \leq x)$ of a Weibull distribution $F(x, \lambda, k)$:

$$P-\text{value} = P(X \leq x) = F(x, \lambda, k) = 1 - \exp\left(-\left(\frac{x}{\lambda}\right)^k\right) \tag{3}$$

Starting with the inclusion of all samples of a cohort to construct the null distribution, we estimated the first parametric set of $\lambda_1$ and $k_1$ in the Weibull distribution by using the fitdistr function in the R package MASS, and performed the Kolmogorov–Smirnov test (KS test) to assess goodness of fit (step A). By calculating KS statistic (i.e., D score), the KS test examines how well the estimated Weibull distribution was fitted with the empirically observed PIN values[29,30]. Then, P-values of all samples were calculated using the function $F(x, \lambda_1, k_1)$ and degraded samples (P-values < 0.02 by default) were excluded in the further round of constructing a null distribution (step B). Iteratively, step A and step B were repeated until a null distribution was reliably generated and converged, when non-degraded samples were used in the construction of null distribution and the best fit (i.e., the smallest D score) was reached. Finally, P-values of all samples were estimated by using the function $F(x, \lambda_{converged}, k_{converged})$ and the resulting values were used to reject samples according to a user defined, objective P-value threshold.

**Downstream analysis and data visualization.** The mKS and iPIS matrices excluding missing values were clustered and visualized separately using the heatmap.2 function in the R package gplots. The order of columns was sorted by the degree of integrity, measured by mRIN and PIN, respectively, from lowest (left) to highest (right). The cophenetic correlation coefficient was used to measure the similarity between two dendrograms. Except for computational workflow and experimental design, analyses and figures in this manuscript were performed and visualized in R. Final figures were all prepared with Adobe Illustrator.

**Reporting summary.** Further information on research design is available in the Nature Research Reporting Summary linked to this article.

## Data availability

RNA-Seq data have been deposited in the Sequence Read Archive with the BioProject number PRJNA414084. Regarding proteomic analysis of the clinical cohort, the SWATH raw data and analyzed data as well as assay libraries are deposited to the ProteomeXchange Consortium via the PRIDE partner repository with the data set identifier PXD007841. Regarding the benchmarking study, the SWATH raw data and

analyzed data as well as assay libraries are deposited to the ProteomeXchange Consortium via the PRIDE partner repository with the data set identifier PXD013622. The source data underlying Figs. 2–4 are provided as a Source Data file. A reporting summary for this Article is available as a Supplementary Information file. All other data supporting the findings of this study are available from the corresponding authors on reasonable request.

## Code availability

To allow for generic evaluation of protein degradation of sample cohorts, PIN is provided as an open-source R package at https://github.com/ProteomicsTools/PIN.

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

## Acknowledgements

We thank Max Frank, Christian Feller, Moritz Heusel, Ludovic Gillet, Evan Williams, and Patrick Pedrioli for discussions, the ETH SIS team for computational support. The group of R.A. was supported by European Research Council (ERC, grant # 616441 DISEASEAVATARS and ERC, grant # 670821 PROTEOMICS4D), SystemcX.ch project PhosphoNet PPM, the Swiss National Science Foundation (SNSF, grant number: 31003A_166435), SystemsX RTD 2012/191 and 2013/156, and PrECISE project from the European Union Horizon 2020 research and innovation programme under grant agreement No 668858. The group of P.J.W. was supported by a SystemsX.ch project (PhosphoNet PPM), a grant provided by the Foundation for Research in Science and the Humanities at the University of Zurich (SWF) and PrECISE project from the European Union Horizon 2020 research and innovation programme under grant agreement No 668858. The group of T.G. was supported by the Westlake Startup Grant, Zhejiang Provincial Natural Science Foundation of China (Grant No. LR19C050001).

## Author contributions

R.A., W.S., T.G., G.R., N.C.T., and P.J.W. conceived the idea. P.J.W., D.R., T.H., C.F. C.P., J.L., N.P. and J.H.R. procured the clinical samples and performed RNA-Seq experiments. T.G. conducted the proteomic experiments. W.S., T.G., L.L., K.C., Q.Z. and A.B. performed SWATH data analysis. N.C.T., U.W. and G.R. analyzed the RNA data. W.S. and N.C.T. performed the protein and mRNA degradation analysis with critical inputs from Y.Z., R.S., J.J., J.M., M.B., B.C., Y.L., G.R. and other authors. W.S. and P.X. performed and analyzed the benchmarking experiment. W.S. and R.A. wrote the manuscript with inputs from all other authors. R.A. and P.J.W. supervised the study.

## Additional information

**Competing interests:** R.A. holds shares of Biognosys AG which operates in the field covered by the article. The research groups of R.A. and T.G. are supported by SCIEX, which provides access to prototype instrumentation, and Pressure Biosciences Inc, which provides access to advanced sample preparation instrumentation. The remaining authors declare no competing interests.

