## [Peer Review File · Nature Communications]

REVIEWERS' COMMENTS:

Reviewer #1 (Remarks to the Author):

The authors have addressed all of my previous questions and comments, which included concerns about the lack of controls vetting the tool and whether the tool just had poor sensitivity or if the tested samples had low degradation. For instance, the authors addressed the first concern extensively, producing a set of gold standard samples by adding varying levels of proteinaseK. The authors verified that the proteins were degraded via SDS-PAGE, and show that the PIN score determined by their methodology correlated to the degree of protein degradation. This was a necessary experiment to perform. The authors then addressed the second concern by performing additional analyses on small cohorts of gastric and breast cancers that had some degraded samples, and showed that there were, in fact, samples that have degraded protein and lower PIN scores.

All in all- the authors have addressed my main concerns.

Reviewer #2 (Remarks to the Author):

The authors have now included an extensive validation of their approach to measure and report protein stability in clinical samples. This includes benchmarking as well as a broader analysis of proteomics samples interrogated using different approaches. Overall the authors have greatly improved the quality of the manuscript and I believe that analysis of protein stability (PIN) will provide a useful tool for future proteomic analyses of clinical samples.

Overall I think the manuscript is of high quality and interest.

Minor comments/suggestions:

Did the authors observe sets of re-occurring non-tryptic peptides (across specific proteins) that were commonly observed in samples with poor PIN? Identifying and reporting such specific peptides, if they exist, would provide the community with an additional resource for targeted analysis on platforms currently unable to conduct DIA/SWATH

As a minor point, I think the authors should soften the statement made on last line, page 15 “...., suggesting that protein degradation would not likely bias biological and clinical conclusion by”. I am not sure there is sufficient evidence for this. The data set underlying this conclusion appears to be of very high integrity (eg <6% with PIN at/below: 0.941) and thus, this statement will, at the best, only apply to data sets of equal or higher quality.

I am also not sure why the authors included the P value when referring to reference number 3 (Romeo et al BMC Biol 2014) in the introduction (line 81 page 4). This seems unnecessary.

Reviewer #3 (Remarks to the Author):

In this manuscript, the authors proposed a method PIN to measure protein degradation from proteomics data. By applying it on a prostate cancer datasets, the authors revealed that the metric is an accurate indicator of protein degradation degree. While I found it useful for sample quality control in proteomic analysis, I have two concerns corresponding to the method section.

The authors adopted Weibull distribution to model the null distribution of PIN (PIN distribution of non-degraded samples) and estimated the parameters of Weibull distribution by iteratively excluding degraded samples detected by the current null distribution. However, the authors did not demonstrate how well the null distribution is fitted or justify the use of Weibull distribution. And the threshold to detect degraded samples in each iteration is not provided. It is critical because all samples were used in the first iteration and algorithm will stop and fail if no samples are detected as degraded.

REVIEWERS' COMMENTS:

Reviewer #1 (Remarks to the Author):

The authors have addressed all of my previous questions and comments, which included concerns about the lack of controls vetting the tool and whether the tool just had poor sensitivity or if the tested samples had low degradation. For instance, the authors addressed the first concern extensively, producing a set of gold standard samples by adding varying levels of proteinaseK. The authors verified that the proteins were degraded via SDS-PAGE, and show that the PIN score determined by their methodology correlated to the degree of protein degradation. This was a necessary experiment to perform. The authors then addressed the second concern by performing additional analyses on small cohorts of gastric and breast cancers that had some degraded samples, and showed that there were, in fact, samples that have degraded protein and lower PIN scores.

All in all- the authors have addressed my main concerns.

> We thank the reviewer for this nice summary for our revision.

Reviewer #2 (Remarks to the Author):

The authors have now included an extensive validation of their approach to measure and report protein stability in clinical samples. This includes benchmarking as well as a broader analysis of proteomics samples interrogated using different approaches. Overall the authors have greatly improved the quality of the manuscript and I believe that analysis of protein stability (PIN) will provide a useful tool for future proteomic analyses of clinical samples.

Overall I think the manuscript is of high quality and interest.

> We thank the reviewer for this nice summary for our revision.

Minor comments/suggestions:

Did the authors observe sets of re-occurring non-tryptic peptides (across specific proteins) that were commonly observed in samples with poor PIN? Identifying and reporting such specific peptides, if they exist, would provide the community with an additional resource for targeted analysis on platforms currently unable to conduct DIA/SWATH

> We agree with the reviewer that it is valuable for the community to describe frequently observed non-tryptic peptides across specific proteins, especially for targeted analysis in DIA/SWATH. We examined the benchmarking study and obtained 105 non-tryptic peptide ions from 79 proteins for which fully tryptic peptides were not detected at all. This means that in the benchmarking study these 79 proteins could be detected only by non-tryptic peptides. Thus, we provide these 105 non-tryptic peptide ions in the format of spectral library, to facilitate downstream DIA/SWATH analysis. They are publicly available on <https://github.com/ProteomicsTools/PIN/data>, and this information has been added into Supplementary Note 1.

As a minor point, I think the authors should soften the statement made on last line, page 15 “....., suggesting that protein degradation would not likely bias biological and clinical conclusion by”. I am not sure there is sufficient evidence for this. The data set underlying this conclusion appears to be of very high integrity (eg <6% with PIN at/below: 0.941) and thus, this statement will, at the best, only apply to data sets of equal or higher quality.

> As the reviewer suggested, we toned down the passage by adding the phrase, “in this set of samples”.

I am also not sure why the authors included the P value when referring to reference number 3 (Romeo et al BMC Biol 2014) in the introduction (line 81 page 4). This seems unnecessary.

> This has been corrected by deleting the P value.

Reviewer #3 (Remarks to the Author):

In this manuscript, the authors proposed a method PIN to measure protein degradation from proteomics data. By applying it on a prostate cancer datasets, the authors revealed that the metric is an accurate indicator of protein degradation degree. While I found it useful for sample quality control in proteomic analysis, I have two concerns corresponding to the method section.

The authors adopted Weibull distribution to model the null distribution of PIN (PIN distribution of non-degraded samples) and estimated the parameters of Weibull distribution by iteratively excluding degraded samples detected by the current null distribution. However, the authors did not demonstrate how well the null distribution is fitted or justify the use of Weibull distribution. And the threshold to detect degraded samples in each iteration is not provided. It is critical because all samples were used in the first iteration and algorithm will stop and fail if no samples are detected as degraded.

> Regarding the first concern, we examined the null distribution empirically by Weibull distribution, Poisson distribution and Normal distribution, when the PIN algorithm had been developed and refined using the benchmarking dataset. As indicated in the main text (Page 20), we “performed the Kolmogorov–Smirnov test (KS test) to assess goodness of fit (step A). By calculating KS statistic (i.e. D score), the KS test examines how well the estimated Weibull distribution was fitted with the empirically observed PIN values.” As shown in the table below, the null distribution of PIN scores was fitted best by the Weibull distribution, judged by D scores. Thus, we decided to use Weibull distribution to model the null distribution.

	D score
Weibull distribution	0.1333
Poisson distribution	0.3927
Normal distribution	0.3074

Regarding the second concern, we added “ P values < 0.02 by default” into the main text, as the threshold to detect degraded samples in each iteration and the default settings in the software.